# Acoustical and behavioral heuristics for fast interactive sound design

Ava Souaille[1], Vincent Lostanlen[1], Mathieu Lagrange[1]*, Nicolas Misdariis[2], Jean-François Petiot[1]

1 Nantes Université, École Centrale Nantes, CNRS, LS2N, UMR 6004, Nantes, France, 2 STMS Ircam–CNRS–SU, Paris, France

* mathieu.lagrange@ls2n.fr

**Data Availability Statement:** All code and data files are available on the following repository: https://github.com/Souaille/fasterinteractive2022.

**Funding:** The author(s) received no specific funding for this work.

## Abstract

During their creative process, designers routinely seek the feedback of end users. Yet, the collection of perceptual judgments is costly and time-consuming, since it involves repeated exposure to the designed object under elementary variations. Thus, considering the practical limits of working with human subjects, randomized protocols in interactive sound design face the risk of inefficiency, in the sense of collecting mostly uninformative judgments. This risk is all the more severe that the initial search space of design variations is vast. In this paper, we propose heuristics for reducing the design space considered during an interactive optimization process. These heuristics operate by using an approximation model, called surrogate model, of the perceptual quantity of interest. As an application, we investigate the design of pleasant and detectable electric vehicle sounds using an interactive genetic algorithm. We compare two types of surrogate models for this task, one based on acoustical descriptors gathered from the literature and the other based on behavioral data. We find that reducing by a factor of up to 64 an original design space of 4096 possible settings with the proposed heuristics reduces the number of iterations of the design process by up to 2 to reach the same performance. The behavioral approach leads to the best improvement of the explored designs overall, while the acoustical approach requires an appropriate choice of acoustical descriptor to be effective. Our approach accelerates the convergence of interactive design. As such, it is particularly suitable to tasks in which exhaustive search is prohibitively slow or expensive.

## 1 Introduction

Sound design may be defined as the creation of sounds under constraints, and aiming at "making an intention audible" in a given context of use [1]. How effectively a sound communicates the designer's intent, depends on how it is experienced by the people it is addressed to. Thus, sound designers can greatly benefit from knowledge in sound perception that is specific to the project they are working on.

Data-informed decision making is an innovative way for addressing this issue, for example through the use of interactive optimization methods. These methods allow users to interactively explore a design space [2], producing data about their choices or behaviors.

**Competing interests:** The authors have declared that no competing interests exist.

## 1.1 Problem statement

However, in this approach, the user's perception of the product is considered as a black-box function, meaning that the explicit relationship between its input (i.e. the designs to evaluate) and its output (i.e. the perceptive evaluations) is not known. The main challenge of optimizing such an unknown function, when it is provided by a human, is the limited number of evaluations of the function that can be requested, before user fatigue makes the harvested evaluations unreliable.

The issue of efficient optimization is thus all the more pressing in the use case discussed in this paper, since we consider the design of an Acoustic Vehicle Alert System (AVAS) for electric or quiet vehicles [3].

Beyond this application setting, expensive measurements or simulations (time-wise and money-wise) can be found in various other fields and are also a strong limitation when optimizing the design of such complex systems. In aerospace vehicle engineering for example [4, 5], precise prediction of system behavior may require several-day long simulations involving multiple types of physical simulations, such as fluid dynamics, chemistry and mechanics. Sometimes, measurements themselves are time consuming, or simply too expensive to be conducted repeatedly.

A common approach to allow for faster data acquisition, is the use of a surrogate model [6, 7], which is an approximation model of the designed system. Such a model can be used, ahead of, or during the optimization process. This model provides some inexpensive estimations of the cost function of the optimization problem. The surrogate model can be used to perform design space reduction [6], by identifying ranges of the design variables that are likely to contain the optimum and performing the optimization in these restricted ranges.

## 1.2 Case study

In this paper, we present several methods for reducing the design space before the interactive optimization process, based on the construction of surrogate models. In the case of a minimization problem, our proposed heuristics aim at reducing the domain of the exploration to a contiguous sub-region in which the cost function is lower, on average. By doing so, we intend to find better solutions during the optimization process. In turn, this allows to reach a given quality of solutions within a reduced number of iterations. Several surrogate modeling approaches are compared, with regards to their cost in terms of human evaluations versus the expected improvement to the optimization process.

We want to test 14 different experimental conditions in total, with a single test taking approximately 40 minutes for one subject, if done by a human. In order to gather enough data within a reasonable time frame, we choose here to numerically simulate the interactive optimization process for the experiments reported in this paper. For this purpose, we build artificial subject models from previous experimental data issued from an experiment that considered the same sound design problem [8].

For every surrogate model considered, the same method is used for reducing the design space.

In this study, the sounds considered were generated using an additive synthesis method controlled using six parameters with four modalities each. The six parameters correspond respectively to: motor/chord proportion, fundamental frequency, harmonic/noise proportion, number of harmonics, amplitude modulation frequency, and amplitude modulation ratio, see [8, 9] for more details. This corresponds to a number of $4^6$ = 4096 possible designs. It is unrealistic to solve this problem using a design of experiments (DOE) approach, thus motivating the

use of an interactive optimization algorithm. Spectrograms of sound examples generated with this synthesis method are shown in Appendix 7.1.

Experiments demonstrate the interest of using a surrogate model based on appropriate behavioral data related to human perception for reducing the design space. It allows us to remove one or two of the less relevant modalities in order to reduce the design by a factor of 5 or 64 respectively. This reduction allows the subject to find better performing solutions than exploring the full design space, when the number of available function evaluations is limited. Alternatively, a surrogate model based on an acoustical descriptor is also proposed to improve the quality of the solutions found through the optimization process, compared to exploring the full design space. Our experiments show that the acoustical approach leads to less consistent improvements than when using a perceptual data based model, even though selected descriptors are considered as relevant in the literature for predicting the perceptual quantities of interest.

### 1.3 Contributions

The contributions and findings of this paper are:

- The introduction of a design space reduction method for interactive optimization, based on surrogate modeling. We focus on the independent minimization of two quantities which likely involve different cognitive processes: namely, unpleasantness and detectability.

- For each of these two quantities, the performance of the optimization process is measured, depending on the type of surrogate model used for the design space reduction. On average, we find that reducing the design space with the tested methods significantly improves the solutions found, until halfway through the maximum number of iterations considered.

- For the two tasks at hand, being the minimization of the perceived unpleasantness and detection time of an electric vehicle warning sound, the behavioral surrogate model is found to lead to a better reduction of the design space, compared to the acoustical surrogate model.

Section 2 describes the proposed surrogate-based design space reduction method. In Section 3, this method is applied to the design of an electric vehicle warning sound. Lastly, Section 4 details the conducted experiments and their results, which are further discussed in section 5.

## 2 Proposed methodology

We want to compare the solutions found through an optimization process when exploring the full design space, to the ones found when the explored design space is reduced beforehand using a surrogate model. This reduction is done by removing modalities from the design variables, which are categorical. We hypothesize that when a limited number of function evaluations is available, reducing the design space allows us to find solutions that outperform the ones found when the full design space is explored.

To evaluate this hypothesis, we apply the design space reduction method using the surrogate model for two interactive optimization problems. The first one is the minimization of the unpleasantness of a designed warning sound. The second one is the maximization of the detectability of an approaching electric vehicle's sound, i.e. minimizing the time it takes for a person to detect an approaching vehicle by its sound.

For both problems, we optimize the design of a sound via an IGA, which works by combining solutions based on subject ratings provided during the optimization process.

We investigate two surrogate modeling strategies and the resulting design space reductions, illustrated in Fig 1:

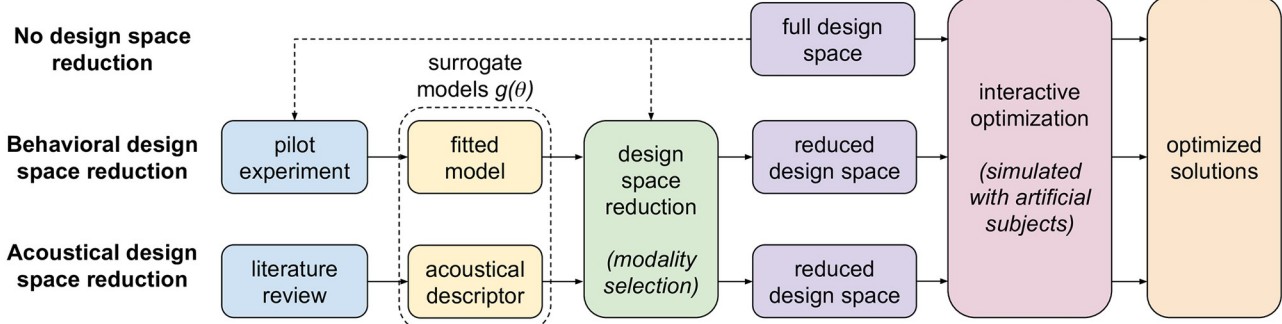

**Fig 1. Evaluated design strategies.** Top: the full design space is explored using an interactive optimization techniques, considered here as a reference. Middle: a pilot experiment allows us to reduce the design space by removing less relevant modalities. Bottom: acoustical descriptors that are assumed to model well the perceptual quantity of interest are used to reduce the design space by removing less relevant modalities.

- **Behavioral Method (BM)**: the surrogate model is fitted from perceptual data, obtained during a pilot experiment, with similar experimental conditions to the interactive optimization experiment.

- **Acoustical Method (AM)**: the surrogate model is an acoustical descriptor that, according to the literature, is assumed to be correlated to objective function.

As shown in Fig 1, we compare the proposed methods based on the results of a subsequent optimization process. The exploration of the full design space with the same optimization algorithm serves as a reference, that we refer to as **Full design space Method (FM)**.

## 2.1 Design space reduction

The interactive optimization problem is to find a sound that minimizes a given perceptual quantity. Let us denote by $f$ this perceptual quantity, also called the cost function of the optimization problem. The design space $D$ is defined by $V$ variables, each of which can take $M_v$ possible values, called modalities:

$$D = \prod_{v=1}^{V} \{1, 2, \cdots, M_v\} \tag{1}$$

A setting is defined as a combination of variable modalities, and denoted by $\theta = (x_1, \cdots, x_V) \in D$. Under a "full factorial" experiment design, the number of possible settings in $D$ is equal to:

$$\prod_{v=1}^{V} M_v = M_1 \times M_2 \times \ldots \times M_V \tag{2}$$

A synthesizer is used to create a sound from each possible setting $\theta$. Solving the optimization problem amounts to finding the setting $\theta$ that minimizes the cost function $f(\theta)$.

In this paper, the goal of the design space reduction method is to reduce the number of modalities per variable $M_v$, so that good solutions are reached faster. We choose to eliminate modalities in order to preserve a structure for which every possible modality combination is a valid setting. The use of the same IGA to explore the initial and the reduced design space is then valid as it works by iteratively recombining the settings of the best solutions. We choose

not to proceed by eliminating settings, because by doing so the solution recombination process of the IGA may lead to out-of-domain modality combinations.

By eliminating modalities, we aim to reduce the full design space $D$ to a subset $D' \subset D$, such that $\bar{f}(D') < \bar{f}(D)$, with $\bar{f}$ being the mean value of the cost function $f$. By doing so, we expect to find solutions with a lower cost function value on average, at least in the beginning of the optimization process.

Reducing the design space $D$ directly based on the value of $f(\theta)$ requires numerous human evaluations. However, the design space reduction is only a preliminary step to a subsequent interactive optimization experiment. Thus, it is important to limit the experimental cost of the proposed method. This is why we propose to use a surrogate model $g(\theta)$ of $f(\theta)$ for our analysis.

## 2.2 Modality selection

For each design variable $x_v$, we want to select a subset of the initial modalities $\varepsilon'_v \subseteq \{1, 2, \cdots, M_v\}$. For this, we consider a setting to be a random variable $\Theta = (X_1, \cdots, X_V)$, following a discrete uniform distribution over the sample space $D = \prod_{v=1}^{V}\{1, 2, \cdots, M_v\}$.

Let us call $G_v(m)$ the conditional expectation of $g(\Theta)$ given $X_v$:

$$G_v(m) = \mathbb{E}[g(X_1, \cdots, X_V) \mid X_v = m] \tag{3}$$

$G_v(m)$ is the expected value of $g(\Theta)$, when $X_v = m$ and can be computed as:

$$G_v(m) \quad = \frac{1}{\prod_{u \neq v} M_u} \sum_{x_1} \cdots \sum_{x_{v-1}} \sum_{x_{v+1}} \cdots \sum_{x_V} g(x_1, \cdots, x_{v-1}, m, x_{v+1}, \cdots, x_V) \tag{4}$$

The proof for Eq 4 can be found in Appendix 7.2.

For a given variable $x_v$, we reduce the number of modalities to $k_v$, with $1 \leq k_v < M_v$, by sorting $G_v(m)$ according to $m$ and eliminating the $M_v - k_v$ modalities with the highest $G_v(m)$ values. Let us call $\Theta'$ the random variable corresponding to the reduced sample space $D' = \prod_{v=1}^{V}\{1, 2, \cdots, k_v\}$. By convention, we number the modalities $m$, so that the values $G_v(m)$ are in increasing order.

By eliminating the modalities with the highest $G_v(m)$ we reduce the expected value of the surrogate model:

$$\mathbb{E}[g(\Theta')] \quad < \mathbb{E}[g(\Theta)] \tag{5}$$

The proof of Eq 5 is in Appendix 7.3.

In this paper, the values of $G_v(m)$ are computed from the expression in Eq 4, by sampling settings over the full design space.

We now have defined how we will reduce the number of modalities per variable, by estimating the conditional expectation $G_v(m)$, based on the surrogate model $g(\theta)$. This method allows us to remove from the design space the modalities that result in the worst values of $g(\theta)$, on average.

## 2.3 Surrogate modeling

The surrogate model $g(\theta)$ is an approximation model of the perceptual quantity to minimize. By using $g(\theta)$ for reducing the design space, our goal is to remove the modalities which also result in the largest values of $f(\theta)$, on average. Thus, it is crucial that $g(\theta)$ provides a good approximation of $f(\theta)$.

For the acoustical approach, the surrogate model is an acoustical descriptor that correlates with the perceptual quantity to minimize, according to the available literature. We choose to compute two descriptors for each cost-function to model, one based on temporal information and one based on spectral information, respectively. While there are studies associating the unpleasantness of a vehicle sound to well known acoustical descriptors, it is not the case for detectability, for which descriptors had to be defined specifically for this study. Thus, we expect the design space reduction to be more consistently adequate across descriptors for unpleasantness than for detection time.

In this case, $g(\theta)$ is obtained by computing the chosen acoustical descriptor for the audio signal resulting from the setting $\theta$.

For the behavioral approach, this model is built from human evaluations of a subset of sounds of the design space. These evaluations are collected during a pilot experiment involving $S$ subjects, with a presentation and evaluation process similar to the one used in the following interactive optimization experiment. During this pilot experiment, each subject $s$ evaluates a subset of $D$ composed of $T$ design settings $h_s(t) = \theta$, for $t \in [1, T]$. We call $f_s(\theta)$ the black-box function corresponding to the evaluation that the individual $s$ would make of the sound corresponding to the setting $\theta$. For a group of subjects, we fit a single surrogate model $g(\theta)$ by minimizing the function

$$g^* = \arg\min_g \sum_s \sum_{t=1}^{T} |f_s(h_s(t)) - g(h_s(t))|^2 \tag{6}$$

The number of optimization variables will depend on the choice of the model $g(\theta)$.

## 2.4 Statistical hypothesis testing

We have presented a method to reduce the design space of settings $\theta$ by evaluating the surrogate function $g(\theta)$, aggregating it by variable (hence $G_v(m)$, see Eq 4), and retaining the modalities $m$ minimizing $G_v(m)$ for each variable $v$. It remains to be seen whether this procedure actually serves our ultimate goal, i.e, to reduce the average value of the cost function $f$. Yet, answering this question precisely would require to collect human judgments for the full design space, and thus defeat the purpose of our heuristics. Instead, we propose to answer it probabilistically: that is, by relying on a correlation coefficient $\rho$ which links $f$ to $g$, and which we assume to be known beforehand.

Let $\mu_f$ (resp. $\mu_g$) be the average values of $f$ (resp. $g$) on the full design space. The Pearson correlation between $f$ and $g$ is defined as

$$\rho = \frac{\sum_\theta (f(\theta) - \mu_f)(g(\theta) - \mu_g)}{\sqrt{\sum_\theta (f(\theta) - \mu_f)^2} \sqrt{\sum_\theta (g(\theta) - \mu_g)^2}} \tag{7}$$

where the sums are taken over the full design space. Despite having no direct access to $f$, we may use the equation above to define a function $\tilde{f}$ whose correlation with $g$ equals $\rho$. Then, we will aggregate values of $\tilde{f}$ per modality $m$ of each variable $v$, yielding a matrix $\tilde{F}_v(m)$. This will allow us to assess whether inequalities in Eq 18 hold consistently for $G_v(m)$ and $\tilde{F}_v(m)$.

In practice, we define $\tilde{f}$ as a noisy version of $g$, where the noise is additive, zero-mean, Gaussian, and independent of setting $\theta$:

$$\tilde{f}(\theta) = g(\theta) + \mathcal{N}(0, \sigma^2). \tag{8}$$

Note that $\mu \tilde{f} = \mu_g$. Plugging the equation above into Eq 7 yields

$$\rho^2 = \frac{\left(\sum_\theta (g(\theta) - \mu_g + \mathcal{N}(0, \sigma^2))(g(\theta) - \mu_g)\right)^2}{\sum_\theta (g(\theta) - \mu_g + \mathcal{N}(0, \sigma^2))^2 \sum_\theta (g(\theta) - \mu_g)^2} \tag{9}$$

We write down our assumption of independence in the finite-sample case as:
$|\sum_\theta \mathcal{N}(0, \sigma^2)(g(\theta) - \mu_g)| \ll \sum_\theta (g(\theta) - \mu_g)^2$. In other words, we assume that there are suffi-ciently many settings in the design space so that the covariance between $g$ and the noise can be neglected. Hence the approximate formula:

$$
\begin{aligned}
\rho^2 &\approx \frac{\left(\sum_\theta (g(\theta) - \mu_g)^2\right)^2}{\left(\sum_\theta (g(\theta) - \mu_g)^2\right)^2 + \left(\sum_\theta \mathcal{N}(0, \sigma^2)^2\right)\left(\sum_\theta (g(\theta) - \mu_g)^2\right)} \\
&\approx \frac{1}{1 + \dfrac{\sum_\theta (\mathcal{N}(0, \sigma^2))^2}{\sum_\theta (g(\theta) - \mu_g)^2}} \approx \frac{1}{1 + \dfrac{\sigma^2}{\sigma_g^2}} \quad ,
\end{aligned}
\tag{10}
$$

where $\sigma_g$ is the standard deviation of $g$ over the full design space. From the above, we derive the standard deviation of additive noise:

$$
\begin{aligned}
\frac{\sigma}{\sigma_g} &= \sqrt{\frac{1}{\rho^2} - 1} \\
\Leftrightarrow \sigma &= \sigma_g \sqrt{\frac{1}{\rho^2} - 1}
\end{aligned}
\tag{11}
$$

$\sigma$ is the standard deviation of the noise added to $g(\theta)$ to compute or estimator $\tilde{f}(\theta)$. If $\rho = 1$, meaning if there is a linear relationship between $f(\theta$ and $g(\theta)$, then we have $\sigma = 0$. In that case, our estimator $\tilde{f}(\theta)$ is directly equal to $g(\theta)$. Knowing $\sigma$ allows us, for each variable and modal-ity, to generate a finite number of random samples of $\tilde{F}_v(m)$. From these samples, we estimate, for each variable, the probability that by discarding the modalities with the highest $G_v(m)$ we do not discard the modalities with the highest $\tilde{F}_v(m)$.

Fig 2 shows an example of the modality selection process based on a surrogate model, together with the probability that the selection operated based on $g$, leads to a wrongful modal-ity elimination for $\tilde{f}$, namely the probability
$P(\tilde{F}_v(n \in \{1, 2, \cdots, k_v\}) > \tilde{F}_v(m \in \{k_v + 1, k_v + 2, \cdots, M_v\}))$. A probability of 0 means that the discarded modalities are always the ones for which $\tilde{F}_v(m)$ have the highest values. A proba-bility of 1 means that there is always at least one kept modality for which $\tilde{F}_v(m)$ is higher than for the discarded modalities. In this example, we want to keep two modalities per variable, using loudness as a surrogate model for unpleasantness, which we want to minimize.

## 3 Application to interactive sound design

In this section, we document our choice of surrogates for the application at hand: interactive sound design of AVAS warning sound.

### 3.1 Behavioral surrogate

The data used for the construction of the behavioral surrogate models was obtained during an interactive multi-objective optimization experiment whose aim is the bi-objective design of an electric vehicle sound presented [8]. This sound should jointly minimize its unpleasantness

## Expected value of Loudness as a surrogate model for Unpleasantness

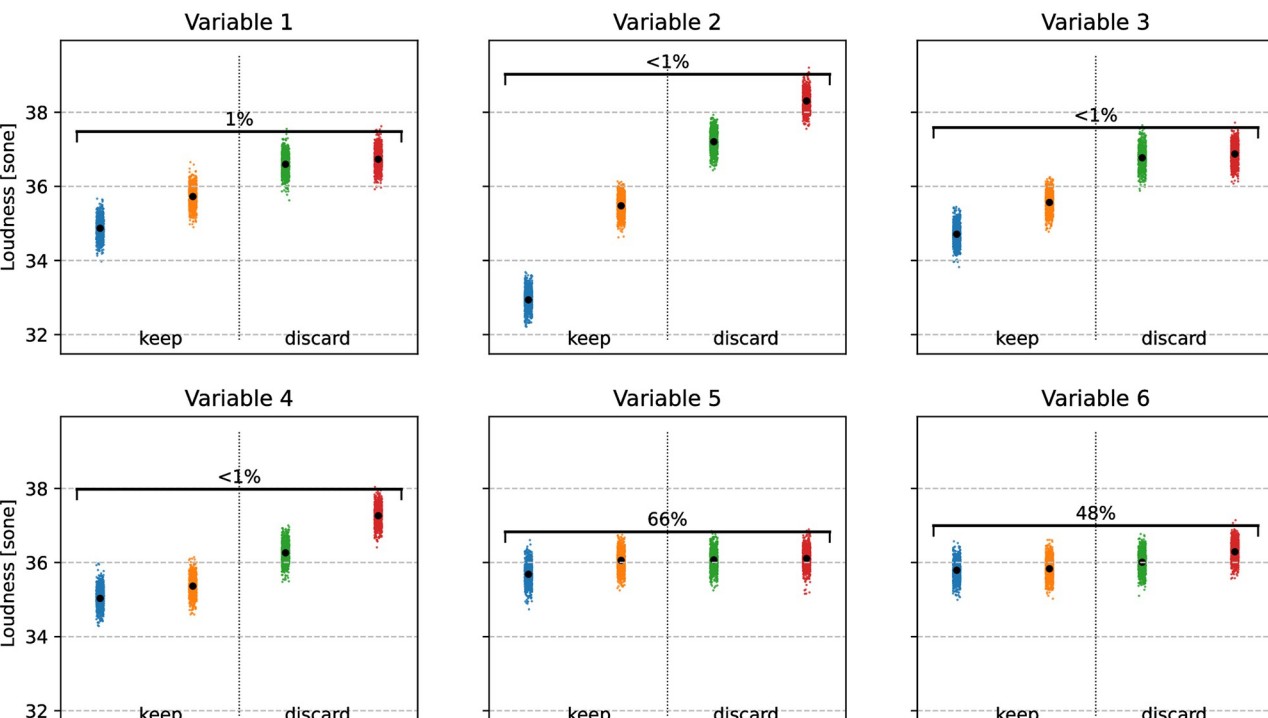

**Fig 2. Conditional expectation $G_\nu(m)$ of the surrogate model $g$ given each design variable.** In this example, there are six design variables, each with four modalities. Here, $g$ is the loudness, used as a surrogate model for the unpleasantness, in order to eliminate two modalities per variable. The colored clusters are random samples of $\tilde{F}_\nu(m)$, with a random horizontal offset for better display. The values of $G_\nu(m)$ are represented by black dots and sorted in increasing order. The percentage above the second and third modalities is the estimated probability that the eliminated modalities are not the ones with the highest $\tilde{F}_\nu(m)$ value, that is $P(\tilde{F}_\nu(n \in \{1, 2, \cdots, k_\nu\}) > \tilde{F}_\nu(m \in \{k_\nu + 1, k_\nu + 2, \cdots, M_\nu\}))$. Sone is a unit of loudness.

and the associated vehicle detection time. A busy street background excerpt is played and the electric vehicle sound is coming randomly either from the left or the right. Each time a vehicle sound is evaluated by a subject, a background extract is randomly selected within a 42 seconds long recording.

In this study, 28 engineering students (15 male, 13 female) performed the experiment. Each one of them made $T = 99$ evaluations, corresponding to 11 generations of 9 sounds. The evaluated sounds were different for each participant and were selected by an IGA during the optimization process. During the test, the subjects listen to a scenario, in which they are standing next to a busy street intersection, with an approaching electric vehicle.

From this data, the models are linear models fitted using the least squares method, according to Eq 6.

### 3.2 Acoustical surrogate

In order to apply a design space reduction based on acoustical descriptors, we need to choose suitable acoustical descriptors for modeling unpleasantness and detection time.

**3.2.1 Unpleasantness.** In [10], the authors find a strong correlation between subjective unpleasantness judgments by 22 subjects and mean Zwicker's loudness (ISO 532 B), with $r = 0.95$. In [11], a perceived unpleasantness model is built from paired comparisons by 60

subjects. Mean roughness is found to be correlated to the model's unpleasantness prediction ($r = 0.97; p < 0.001$). A significant correlation ($r = 0.71; p = 0.021$) is also found with median loudness (N50). In [12], annoyance ratings for 7 categories of vehicles are provided by 30 subjects. Each category is a combination of one or more vehicle types (among bus, two-wheeled, light and heavy vehicles) and a driving condition (acceleration, constant speed and deceleration). For 5 out of 7 of the categories, Zwicker's loudness is the factor that contributes the most to annoyance. For the category "Two wheeled vehicles in acceleration", the loudness, integrated between 15 and 18 Barks has a higher correlation to annoyance than total loudness ($r = 0.81; p < 0.001$ and $r = 0.72; p < 0.001$, respectively). For the category "Light vehicles in acceleration", annoyance is mostly correlated to $L_{Aeq}$ computed between the one-third octave bands between 315Hz and 1250Hz. In [13], 19 subjects rated the annoyance of 4 vehicle pass-by scenarios. A high correlation is found with mean loudness ($r = 0.96$), roughness ($r = 0.97$) and fluctuation strength ($r = 0.93$).

From this review, we choose to select the Zwicker's loudness and Daniel and Weber's roughness. The Zwicker's loudness is computed using the python module AudioCommons Timbral Models, which implements the algorithm from [14]. Roughness is computed using Daniel and Weber's model [15], with the python module MoSQITo. as approximated unpleasantness models. The assumption is that sounds that are not unpleasant tend to have a lower loudness and/or roughness. We call the reduction method based on the loudness descriptor $AM - L$ and the one based on the roughness descriptor $AM - R$.

**3.2.2 Detection time.** In [16], auditory salience is studied through the paired evaluations of 20 sound scenes, using a dichotic listening paradigm. The 50 subjects had to indicate in real time which scene attracted their attention (left or right). From all the subject panel's evaluations, salient events are detected and their salience level is estimated. Several acoustical metrics are computed, as well as more complex models, fitted on the experimental data. The authors observe a correlation between the salience level of an event and the loudness change preceding the event ($r = 0.44$, $p < 1.0 \times 10^{-23}$). They also remark that the reaction time of the subjects is shorter for events with a high salience level.

Several international organizations such as the United Nations [17] and the European Union [18] require the electric vehicles to be equipped with an AVAS warning sound, to indicate their presence to other road users, such as pedestrians and cyclists. In particular, the United Nations require a minimum sound pressure level (in dBA) per one-third octave band, at 10 and 20 km/h. This system ensures that the vehicle sound emerges from the background noise. Following this idea, we choose to build a metric based on the power difference between the vehicle and the background, integrated over several frequency bands.

We thus consider two metrics: the loudness change before the vehicle's passing-by, as well as the acoustical power difference per frequency band. We assume that for these two metrics, the higher the value, the lower the detection time. We call the reduction method base on the loudness difference descriptor $AM - dL$ and the one based on the power difference $AM - dP$.

In [16], loudness change is computed as the mean loudness during the time window going from -2 seconds to -1.5 seconds before the salient event (window **f1**), subtracted from the mean loudness during the 0.5 seconds preceding the event (window **f2**). Fig 3 shows the two time windows. The salient event is defined as the moment the vehicle passes in front of the subject.

The power of the spectrogram of the vehicle passing-by is averaged over time, to obtain the mean power spectrum of the signal. The python module Librosa is used to compute the mel-scaled spectrogram, with 128 Mel bands between 0 and 24kHz. The same is done for a background extract of the same duration and the power difference between the vehicle and the background is computed for each mel frequency band. Finally, the power differences are

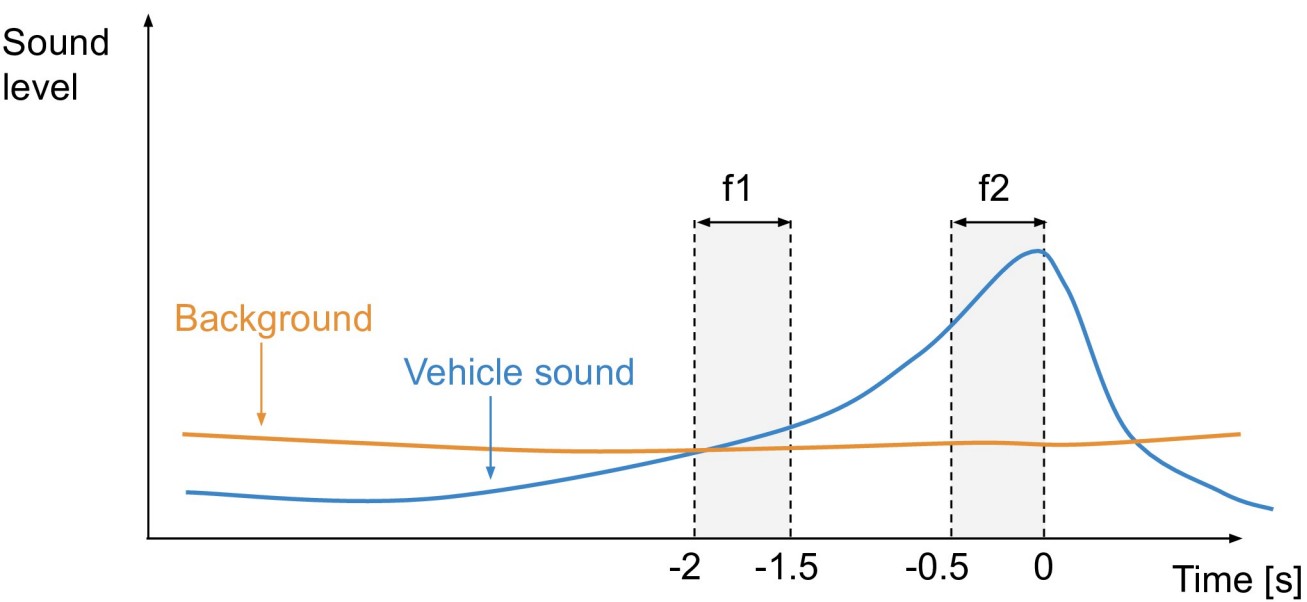

**Fig 3. Time windows used to loudness change calculation, shown in gray.**

summed over all the frequency bands for which the vehicle signal's power is higher than the background's.

To take into account the random background choice in the initial experiment, all acoustical metrics described in this Section are computed by combining the vehicle sounds with several background excerpts, one at a time, and then averaging the metric value over all the sound/background combinations. In order to predict the conditional means described in Section 2.2, the descriptors are computed for every possible design in the design space (4096 in total).

### 3.3 Statistical relevance of surrogates

Based on the method described in section 2.4, we estimate the probability of making a wrongful modality elimination with each surrogate, when eliminating two modalities per variable. Using our dataset, we estimate the correlation coefficients between our surrogates and our available human judgments. The results are shown in Tables 1 and 2 for unpleasantness and detectability, respectively. The probabilities are estimated using a Monte-Carlo method, by generating 1000 samples of $\tilde{f}(\theta)$ for each surrogate and each possible setting $\theta$.

The probabilities of making a wrongful modality elimination are very low for most variables and most surrogate models. These probabilities depend on the correlation coefficient between the surrogate models and the perceptual quantities to approximate, but also on the number of

**Table 1. Estimated probabilities of making a mistake in the modality selection based on *g*, in the case of unpleasantness.** These are the results when eliminating two modalities per variable out of four. **AM-L**: acoustical method with loudness, **AM-R**: acoustical method with roughness, **BM**: behavioral method.

| ρ from our experimental data | | | | | | | |
|---|---|---|---|---|---|---|---|
| $\rho$ | Model | $x_1$ | $x_2$ | $x_3$ | $x_4$ | $x_5$ | $x_6$ |
| 0.32 | **AM-L** | 0.01 | 0.00 | 0.00 | 0.01 | 0.66 | 0.48 |
| 0.35 | **AM-R** | 0.00 | 0.00 | 0.21 | 0.34 | 0.39 | 0.36 |
| 0.51 | **BM** | 0.00 | 0.00 | 0.00 | 0.00 | 0.00 | 0.00 |

**Table 2. Estimated probabilities of making a mistake in the modality selection based on $g$ leads to at least one wrongful elimination, in the case of detection time.** These are the results when eliminating two modalities per variable out of four. **AM-dL**: acoustical method with loudness difference, **AM-dP**: acoustical method with power difference, **BM**: behavioral method.

| $\rho$ from our experimental data | | | | | | | |
|---|---|---|---|---|---|---|---|
| $\rho$ | Model | $x_1$ | $x_2$ | $x_3$ | $x_4$ | $x_5$ | $x_6$ |
| -0.25 | **AM-dL** | 0.19 | 0.00 | 0.01 | 0.07 | 0.51 | 0.20 |
| -0.38 | **AM-dP** | 0.31 | 0.03 | 0.00 | 0.07 | 0.59 | 0.00 |
| 0.49 | **BM** | 0.00 | 0.00 | 0.00 | 0.00 | 0.24 | 0.00 |

points used to estimate each value of $\tilde{F}_v(m)$. This number results from the parametrization of the problem, meaning the number of variables and the number of modalities per variable. The low probabilities of error in Tables 1 and 2 comfort us in the belief that, given the correlation between the chosen models and the quantities to minimize, as well as the initial parametrization of the problem, our heuristics can perform an adequate reduction of the design space and have, on average, a lower value of $f$ for $D'$ than for $D$.

# 4 Results

The implementation of the experimental protocol as well as the datasets generated and/or analysed during the current study are available online: https://github.com/Souaille/fasterinteractive2022.

The interactive optimization experiment is simulated by building $S = 28$ subject models from the dataset described in Section 3.1, one for each subject that participated in the pilot experiment. The models are linear, without interactions, with 19 parameters to estimate (the intercept, plus three parameters per design variable). From the scores provided by each of these models as a subject's response, we are able to simulate the IGA process several times per subject. Each time a simulation is made with a subject model, the surrogate model used for the behavioral design space reduction is trained on the data from all other subjects (similar to cross-validation). Even though the subject models are linear, in reality, a subject's perception might be more complex, with interactions between the design variables. Thus, the simulated optimization performances resulting from a reduction with the **BM** method, will mostly depend on the agreement between the subjects from our dataset regarding their evaluations of unpleasantness and detection time.

Since the IGA is a stochastic process, we use the Monte Carlo method to estimate the unpleasantness and detectability of the explored solutions. For each surrogate and each subject model, 50 repeated simulations of optimization with an IGA are done, for 11 generations of 9 sounds. The genetic algorithm uses a mutation rate of 0.05, 2 elites per generation, binary tournament selection and uniform crossover. This makes 99 evaluations for a given subject model during each simulated experiment, which is the same as during the experiment from our dataset. For each simulation, the minimum unpleasantness/detection time value obtained at each generation is selected. These values are then averaged over the 50 simulations. Table 3 summarizes the different experiments that were conducted.

## 4.1 Unpleasantness surrogate

Fig 4 shows the average simulation results for every case described in Table 3. Each data point is the average value over all simulations (28 virtual subjects x 50 simulations per subject) of the smallest unpleasantness value obtained at each generation. The horizontal axis shows the

**Table 3. Simulation summary for the minimization of a) unpleasantness and b) detection time.** $M_v$ is the number of modalities per factor.

| Method | Surrogate model | $M_v$ | Prerequisite |
|---|---|---|---|
| Full design space (**FM**) | None | 4 | None |
| Acoustical (**AM**) | a) Loudness / roughness | 2 or 3 | None |
| | b) Loudness change / power difference | | |
| Behavioral (**BM**) | a) Unpleasantness | 2 or 3 | Pilot |
| | b) Detection time | | |

number of generations, as well as the number of cost-function evaluations that were required to reach each point (number of generations times the number of individual in each generation (9)).

This figure shows that the smaller the design space, the smaller the difference between the minimum unpleasantness value at the first generation and the value at the last generation. What is more, when only two modalities are kept, there is little improvement in the minimum unpleasantness after the seventh generation.

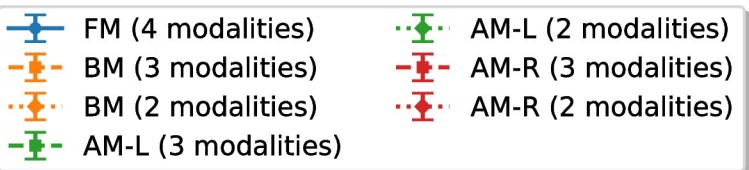

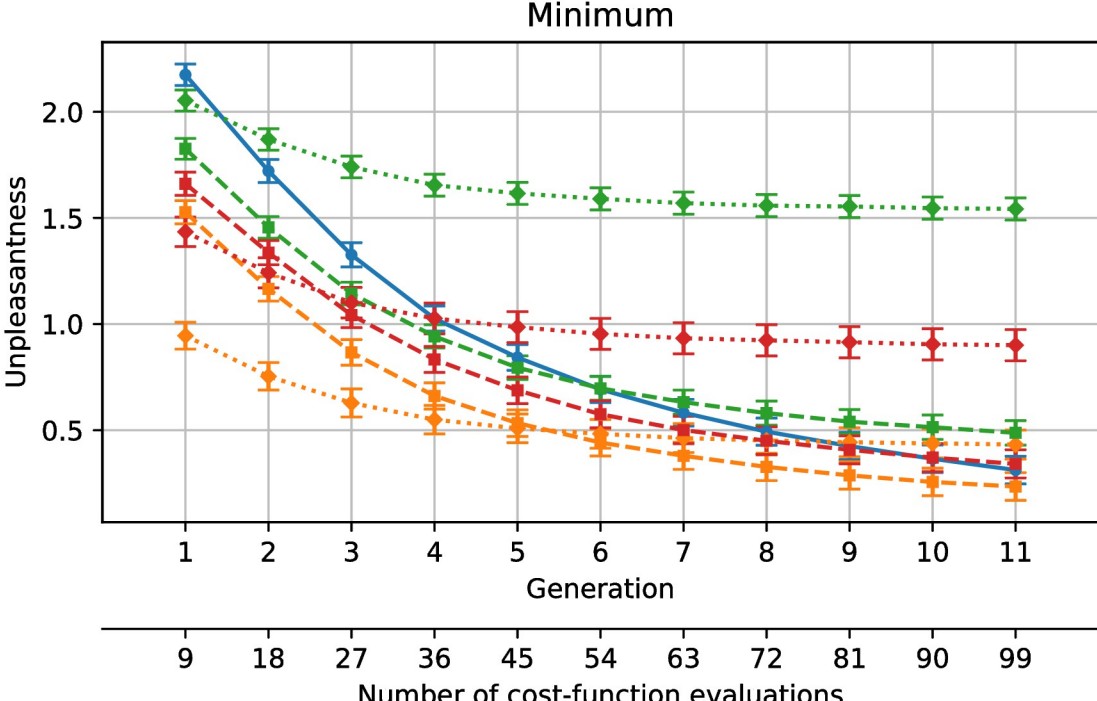

**Fig 4. Minimum unpleasantness value at each generation, averaged over every simulations.** The error bars represent the standard error of the mean. **FM**: full design space method, **AM-L**: acoustical method with loudness, **AM-R**: acoustical method with roughness, **BM**: behavioral method.

**Table 4.** Generations **until which** the design space reduction methods result in significantly ($p < 0.05$) **lower** unpleasantness than without reduction. **AM-L**: acoustical method with loudness, **AM-R**: acoustical method with roughness, **BM**: behavioral method.

| | Number of modalities per variable | |
|---|---|---|
| | **3 modalities** | **2 modalities** |
| **AM-L** | 5 | 1 |
| **AM-R** | 4 | 3 |
| **BM** | 9 | 6 |

A Shapiro-Wilk test indicates that none of the data points was normally distributed, with a significance level of $\alpha = 0.05$. Thus, in order to compare the performance of the methods, we use non-parametric tests. Specifically, we use one-sided Mann-Whitney U tests to compare the unpleasantness of the best solutions found when exploring the full design space, to the unpleasantness of those found after applying each design space reduction method. We perform a test for each generation and look at the last generation at which each method outperforms the **FM** method, shown in Table 4.

The results show that reducing the design space allows for the minimum unpleasantness to be lower at the first generations, for all surrogate models. The solutions are improved for a smaller number of generations, when keeping 2 modalities per variable instead of 3. This shows that because the surrogate models are only approximate models of unpleasantness, it might be risky to reduce the design space too much based on their predictions. Reducing the design space with the behavioral method improves the solutions for the greatest number of generations, regardless of the number of modalities that are kept. This is expected, as the surrogate model used in the behavioral method should provide the best approximation of the cost-function. When 3 modalities are kept per variable, the behavioral method outperforms the reference until the 9th generation, which is almost the maximum number of generations for the experiment.

Because the reduction in unpleasantness during the optimization process is smaller for smaller design spaces, exploring the full design space eventually leads to better solutions, if the algorithm is run for enough generations. The generation number at which this happens depends on the surrogate model and is shown in Table 5.

For the acoustical method, the solutions found when reducing to two modalities per variables are outperformed by the reference, after 3 generations for loudness and 6 generations for roughness. None of the other methods is outperformed within the 11 generations of the experiment. To summarize our study on unpleasantness, reducing the design space allows us to find solutions that are at least equivalent to those found without prior reduction, in most cases studied here. Similarly, it allows to reach solutions with a given unpleasantness value in a reduced number of iterations.

**Table 5.** Generations **from which** the design space reduction methods result in significantly ($p < 0.05$) **higher** unpleasantness than without reduction. **AM-L**: acoustical method with loudness, **AM-R**: acoustical method with roughness, **BM**: behavioral method.

| | Number of modalities per variable | |
|---|---|---|
| | **3 modalities** | **2 modalities** |
| **AM-L** | - | 3 |
| **AM-R** | - | 6 |
| **BM** | - | - |

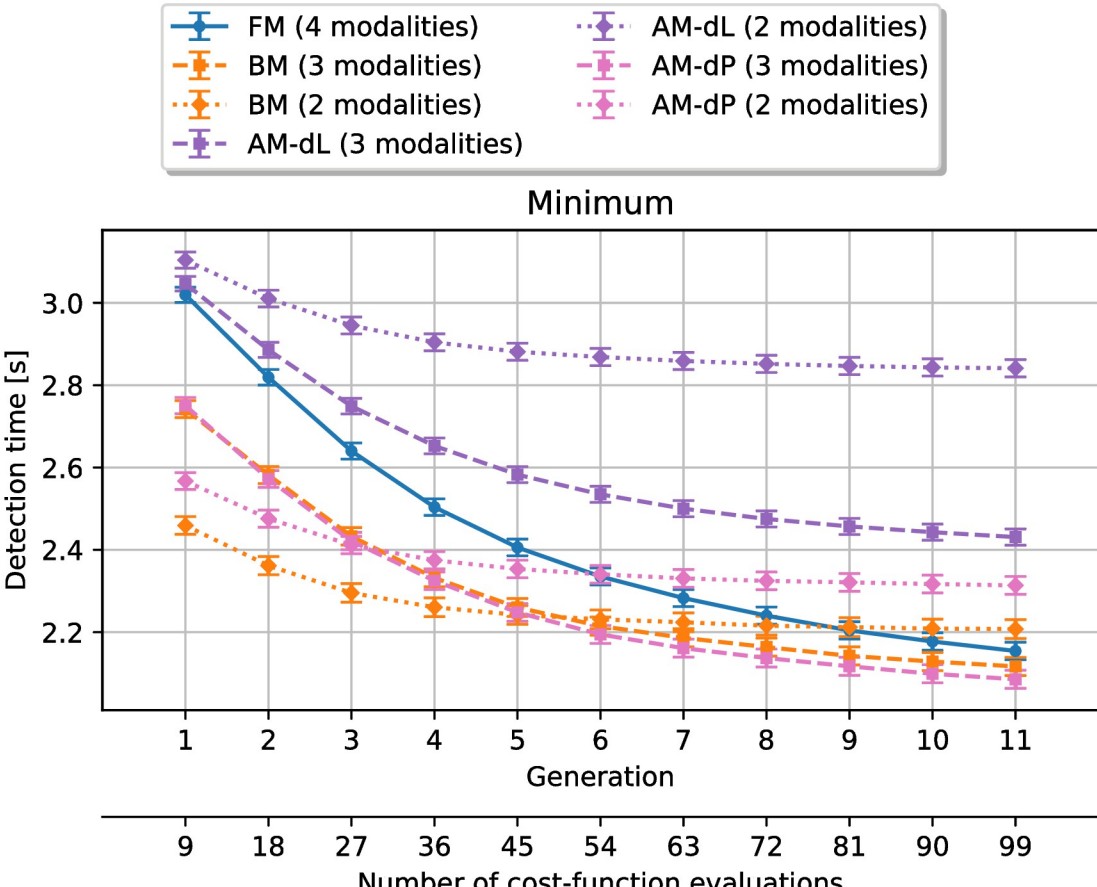

**Fig 5. Minimum detection time value at each generation, averaged over every simulations.** The error bars represent the standard error of the mean. **FM**: full design space method, **AM-dL**: acoustical method with loudness difference, **AM-dP**: acoustical method with power difference, **BM**: behavioral method.

### 4.2 Detection time surrogate

Fig 5 shows the average simulation results for the detection time. Similarly to Fig 4, the difference between the minimum detection time at the beginning and the one at the end of the optimization process is smaller for smaller design spaces.

One can see from the figure that the descriptor based on loudness difference before the arrival of the vehicle (method **AM-dL**) was not a proper surrogate for detection time. Indeed, the corresponding design space reduction leads to results that are worse than the **FM** method, regardless of the number of modalities that are kept. At first glance, the behavioral method (**BM** in the figure) seems to perform similarly as in the case of unpleasantness, with results better or equivalent to the reference. The performances with the second descriptor (power difference, method **AM-dP**) seem on par with the performances for the behavioral method.

Once again, a Shapiro-Wilk test indicates that none of the data points were normally distributed, with a significance level of $\alpha = 0.05$, so one-sided Mann-Whitney U tests are used to compare the methods with the reference at each generation.

Table 6 shows the generation until which each method leads to significantly lower detection time than the **FM** method. The results confirm the observations from Fig 5. The acoustical method with the descriptor based on loudness difference (**AM-dL**) never outperforms the

**Table 6.** Generations **until which** the design space reduction methods result in significantly ($p < 0.05$) **lower** detection time than without reduction. **AM-dL**: acoustical method with loudness difference, **AM-dP**: acoustical method with power difference, **BM**: behavioral method.

|  | Number of modalities per variable | |
| --- | --- | --- |
|  | **3 modalities** | **2 modalities** |
| **AM-dL** | - | - |
| **AM-dP** | 11 | 5 |
| **BM** | 10 | 7 |

reference. The acoustical method with the descriptor based on power difference (**AM-dP**) is on par with the behavioral method. Both of these have even better performances than in the unpleasantness case, especially when keeping three modalities per variable.

Table 7 shows the generation from which each of these methods performs worse than the **FM** method. The case **AM-dL** is quickly outperformed by the reference (from the second generation when keeping three modalities and from the first generation when keeping one modality). When keeping three modalities, the method **AM-dP** leads to results on par with the reference until the last generation. However, the behavioral method still leads to the best results, as it is on par with the reference until the last generation, both when keeping three and two modalities per variable.

## 5 Discussion

In this paper, we proposed heuristics for reducing a design space based on information from a surrogate model. We compared the effects of different surrogate models on an interactive optimization process subsequent to the design space reduction. We observed that if the surrogate model is chosen judiciously, the proposed heuristics lead to an improvement of the solutions found during the optimization process. It could be particularly interesting to include these heuristics in experimental protocols for which the number of available human evaluations is limited.

### 5.1 Choice of surrogate

The experiments show that the choice of the surrogate model is crucial in order to reduce the design space appropriately.

For three out of four of the chosen descriptors, the solutions found during the IGA were not outperformed by the **FM** method within the maximum number of generations, when keeping three modalities per variable. When keeping two modalities per variable, the **FM** method ends up performing better within the 11 generations. However, for one of the descriptors, both reductions were quickly outperformed by the reference. Hence, the IGA process can benefit from a reduction of the design space only using a suitable acoustical descriptor.

**Table 7.** Generations **from which** the design space reduction methods result in significantly ($p < 0.05$) **higher** detection time than without reduction. **AM-dL**: acoustical method with loudness difference, **AM-dP**: acoustical method with power difference, **BM**: behavioral method.

|  | Number of modalities per variable | |
| --- | --- | --- |
|  | **3 modalities** | **2 modalities** |
| **AM-dL** | 2 | 1 |
| **AM-dP** | - | 8 |
| **BM** | - | - |

For unpleasantness, in the best case studied here, that is when keeping three modalities per variable, the behavioral method allows us to find better solutions than without design space reduction, up to the ninth generation (corresponding to 81 evaluations). The improvement is even better for detection time, as the behavioral method allows to find better solutions up to the tenth generation (90 evaluations). In both cases, the **FM** method does not outperform the behavioral method with the maximum number of generations for the experiment.

## 5.2 Benefits of design space reduction

Considering that most of the improvement of the solutions' quality happens during the first generations, regardless of the method, the proposed behavioral space reduction method provides a clear advantage over the exploration of the full design space. Fitting a simple linear model without interactions does not require a large amount of evaluations, especially if the experimental design is chosen accordingly. For example, an optimal experimental design allows to build a simple model from an optimized number of human evaluations. Because of this, it can conveniently be integrated ahead of an interactive optimization experiment.

Exploring the reduced design space with a smaller number of iterations could also allow to have more subjects to participate to the interactive optimization process, as each experiment would be shorter.

That being said, fitting the surrogate model also has an experimental cost. The experiments conducted during this study were only simulated, but in practice a choice should be made regarding the amount of resources allocated to the surrogate modeling, compared to the actual optimization process.

Also, the subject models used for the simulations were linear models without interactions. A real person's behavior is likely to be more complex than that, while also presenting some variability. This would of course influence the results of such a method in a real-life application.

## 5.3 Applicability to other use cases

The method discussed in the paper aims to reduce the design space ahead of an interactive perceptual optimization experiment. In the literature, as in the present study, this type of experiment is often performed with an Interactive Genetic Algorithm (IGA) experiment, but it can be considered prior any iterative design process involving perceptual evaluations.

We review here some use cases tackled with IGAs where the proposed method could readily be applied: design of automobile parts, like dashboards [19], steering wheels [20] and exterior shape of the vehicle [21]; clothing design [22–25]; design of furniture like glasses [2], bottles [26], vases [27] or chairs [28]. Each time, the aim is to find designs that best respond to a given semantic description (for example, a "sporty" dashboard). The subjects concern graphic models of the object to be designed, through an interface. The only constraints is that, given the principle of the proposed reduction method, the design must be defined by a given number of qualitative or quantitative parameters with discrete values only.

## 6 Conclusion

This study demonstrated the potential of using surrogate modeling for design space reduction, in the framework of interactive optimization for sound design purposes.

By means of numerical simulations, we compared two alternative methods for building the surrogate model. One is based on acoustical descriptors found in the literature and the other consists in fitting a model on behavioral experimental data.

The numerical experiments demonstrated that the behavioral approach leads to better results than the acoustical one, for which there is a greater risk of having a surrogate model

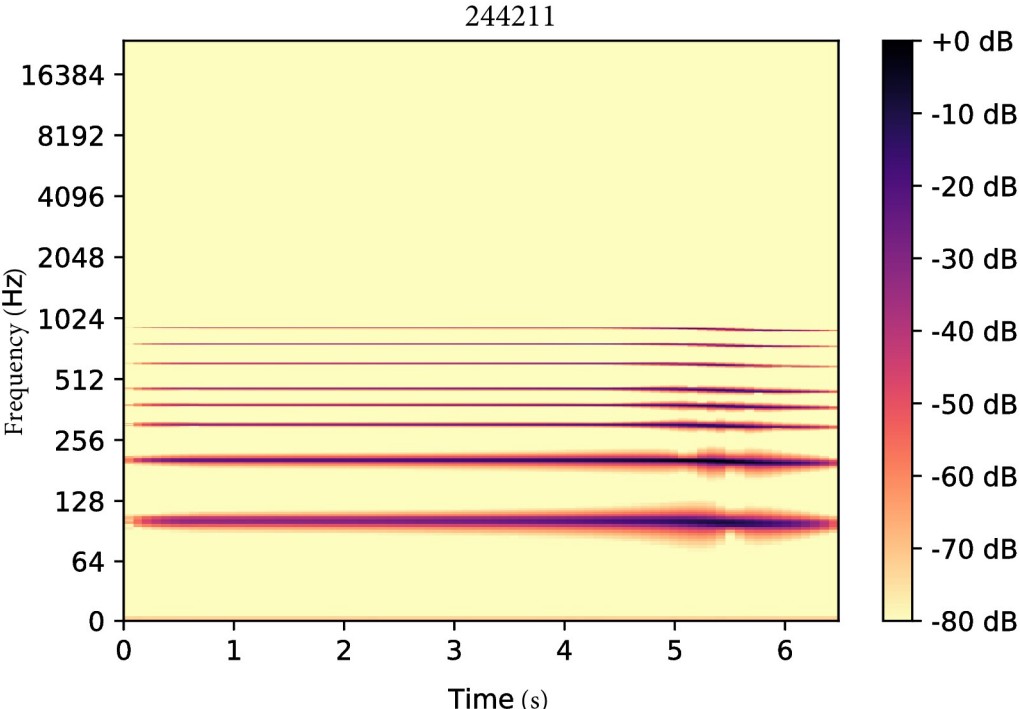

**Fig 6. Fourier spectrogram of the sound generated with the synthesizer with parameters $\theta$ = (2, 4, 4, 2, 1, 1).**

that is not representative of the cost function, if it was not chosen properly. These experiments showed the interest of reducing the design space, in order to reach better solutions when the maximum number of available evaluations is limited. On average, the informed reduction of the 4096 solutions design space of a factor of 64 lead to a reduction of the number of iterations of the design process by a factor of two.

We discussed the limitations of the present study and considerations that should be taken into account when applying the proposed method to a real-life problem. In particular, the benefits and the costs of conducting a pre-experiment, in order to build the surrogate model, should be considered. A compromise should thus be found between the resources allocated to building the model and the risk of reducing the design space to an inadequate subspace.

## 7 Appendix

### 7.1 Spectrograms of sound examples

The synthesizer used in this study generates sounds depending on a set of discrete parameters. Fig 6 plots the Fourier spectrogram of the sound generated with the synthesizer with parameters $\theta$ = (2, 4, 4, 2, 1, 1). Changing one variable at a time can lead to large changes in the sound's spectral properties, as illustrated by Fig 7.

### 7.2 Proof of computation of $G_v(m)$

We consider a setting to be a random variable $\Theta = (X_1, \cdots, X_V)$, following a uniform distribution over the sample space $D = \prod_{v=1}^{V} [\![1; M_v]\!]$. Its probability mass function has a constant

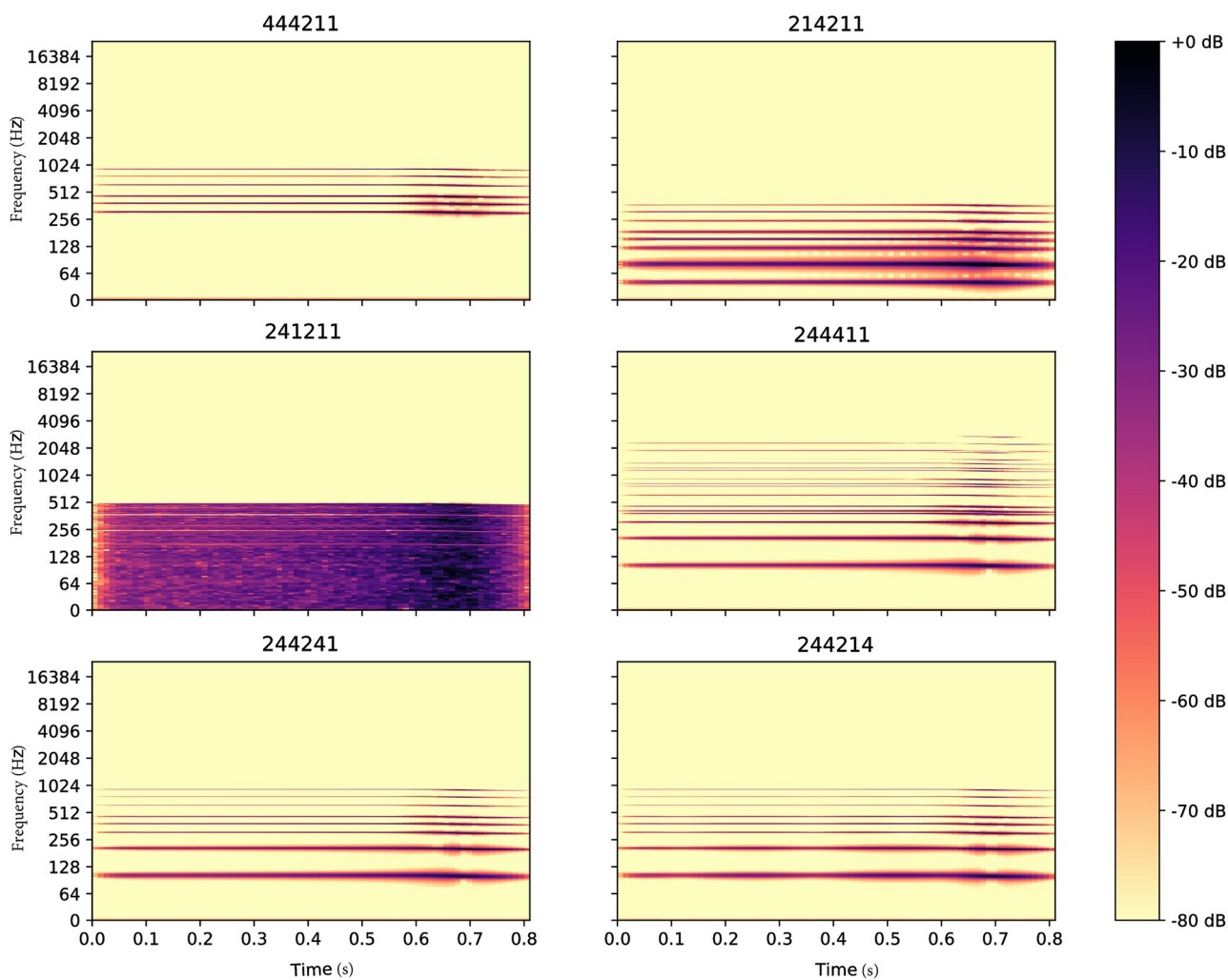

**Fig 7. Fourier spectrogram of the sounds generated with the synthesizer by changing the modality of one variable at a time to another modality, with θ = (2, 4, 4, 2, 1, 1) as a starting point.**

value of:

$$f_{\Theta}(\theta) = \frac{1}{\prod_{u=1}^{V} M_u} \quad (12)$$

Let us call $G_v(m)$ the conditional expectation of $g(\Theta)$ given $X_v$:

$$G_v(m) = \mathbb{E}[g(X_1, \cdots, X_V) \mid X_v = m] \quad (13)$$

Let $Y = g(\Theta)$. $G_v(m)$ can be expressed as:

$$\begin{aligned} G_v(m) &= \sum_y y \mathbb{P}[g(X_1, \cdots, X_V) = y \mid X_v = m] \\ &= \sum_y y \frac{\mathbb{P}[(g(X_1, \cdots, X_V) = y) \cap (X_v = m)]}{\mathbb{P}[X_v = m]} \end{aligned} \quad (14)$$

The settings have a uniform distribution, so $\mathbb{P}[X_v = m] = \frac{1}{M_v}$. Hence:

$$G_v(m) \quad = M_v \sum_y y \mathbb{P}[(g(X_1, \cdots, X_V) = y) \cap (X_v = m)] \tag{15}$$

The probability on the right-handed side can be expressed from the probability mass function $f_\Theta(\theta)$:

$$\begin{aligned} G_v(m) \quad &= M_v \sum_y y \sum_{\theta s.t. (g(\theta)=y) \cap (X_v=m)} f_\Theta(\theta) \\ &= M_v \sum_{\theta s.t. X_v=m} g(\theta) f_\Theta(\theta) \end{aligned} \tag{16}$$

Finally, from 12 we have:

$$\begin{aligned} G_v(m) \quad &= \frac{M_v}{\prod_{u=1}^V M_u} \sum_{\theta s.t. X_v=m} g(\theta) \\ &= \frac{1}{\prod_{u \neq v} M_u} \sum_{\theta s.t. X_v=m} g(\theta) \\ &= \frac{1}{\prod_{u \neq v} M_u} \sum_{x_1} \cdots \sum_{x_{v-1}} \sum_{x_{v+1}} \cdots \sum_{x_V} g(x_1, \cdots, x_{v-1}, m, x_{v+1}, \cdots, x_V) \end{aligned} \tag{17}$$

## 7.3 Proof of $\mathbb{E}[g(\Theta')] < \mathbb{E}[g(\Theta)]$

For a given variable $x_v$, we reduce the number of modalities to $k_v$, with $1 \leq k_v < M_v$, by sorting $G_v(m)$ according to $m$ and eliminating the $M_v - k_v$ modalities with the highest $G_v(m)$ values. Let us call $\Theta'$ the random variable corresponding to the reduced sample space $D' = \prod_{v=1}^V [\![1; k_v]\!]$. By convention, we number the modalities $m$, so that the values $G_v(m)$ are in increasing order:

$$G_v(n) < G_v(m), \forall n < m \tag{18}$$

The expected value of $Y = g(\Theta)$ is:

$$\begin{aligned} \mathbb{E}[g(\Theta)] \quad &= \mathbb{E}[\mathbb{E}[g(\Theta) \mid X_v]] \\ &= \sum_{m=1}^{M_v} \mathbb{P}[X_v = m] \mathbb{E}[g(\Theta) \mid X_v = m] \\ &= \frac{1}{M_v} \sum_{m=1}^{M_v} \mathbb{E}[g(\Theta) \mid X_v = m] \\ &= \frac{1}{M_v} \sum_{m=1}^{M_v} G_v(m) \end{aligned} \tag{19}$$

Similarly, the expected value of $Y' = g(\Theta')$ is:

$$\mathbb{E}[g(\Theta')] = \frac{1}{k_v} \sum_{m=1}^{k_v} G_v(m) \tag{20}$$

From [18], we have:

$$\sum_{n=1}^{k_\nu} \frac{G_\nu(n)}{k_\nu} \;<\; \sum_{m=k_\nu+1}^{M_\nu} \frac{G_\nu(m)}{M_\nu - k_\nu}$$

$$\Leftrightarrow \frac{M_\nu - k_\nu}{M_\nu} \sum_{n=1}^{k_\nu} \frac{G_\nu(n)}{k_\nu} \;<\; \sum_{m=k_\nu+1}^{M_\nu} \frac{G_\nu(m)}{M_\nu} \qquad (21)$$

$$\Leftrightarrow \sum_{n=1}^{k_\nu} \frac{G_\nu(n)}{k_\nu} \;<\; \sum_{m=1}^{M_\nu} \frac{G_\nu(m)}{M_\nu}$$

$$\Leftrightarrow \mathbb{E}[g(\Theta')] \;<\; \mathbb{E}[g(\Theta)]$$

## Author Contributions

**Conceptualization:** Mathieu Lagrange, Nicolas Misdariis, Jean-François Petiot.

**Data curation:** Ava Souaille.

**Investigation:** Vincent Lostanlen, Mathieu Lagrange.

**Methodology:** Ava Souaille, Vincent Lostanlen, Mathieu Lagrange, Jean-François Petiot.

**Project administration:** Jean-François Petiot.

**Software:** Ava Souaille, Mathieu Lagrange.

**Supervision:** Mathieu Lagrange, Nicolas Misdariis, Jean-François Petiot.

**Validation:** Mathieu Lagrange.

**Writing – original draft:** Ava Souaille, Vincent Lostanlen, Mathieu Lagrange, Nicolas Misdariis, Jean-François Petiot.

**Writing – review & editing:** Ava Souaille, Vincent Lostanlen, Mathieu Lagrange, Nicolas Misdariis, Jean-François Petiot.

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
