## [Decision Letter · Decision Letter 0]

14 Sep 2023

PONE-D-23-15140Acoustical and Behavioral Heuristics for Fast Interactive Sound DesignPLOS ONE

Dear Dr. Lagrange,

Thank you for submitting your manuscript to PLOS ONE. After careful consideration, we feel that it has merit but does not fully meet PLOS ONE’s publication criteria as it currently stands. Therefore, we invite you to submit a revised version of the manuscript that addresses the points raised during the review process, especially increasing the discussion about the applicability of the method proposed to different fields.

We look forward to receiving your revised manuscript.

Kind regards,

Sonsoles López-Pernas

Academic Editor

PLOS ONE

Journal Requirements:

Reviewers' comments:

Reviewer's Responses to Questions

**Comments to the Author**

1. Is the manuscript technically sound, and do the data support the conclusions?

Reviewer #1: Yes

Reviewer #2: Yes

2. Has the statistical analysis been performed appropriately and rigorously? 

Reviewer #1: Yes

Reviewer #2: Yes

3. Have the authors made all data underlying the findings in their manuscript fully available?

Reviewer #1: Yes

Reviewer #2: Yes

4. Is the manuscript presented in an intelligible fashion and written in standard English?

Reviewer #1: Yes

Reviewer #2: Yes

5. Review Comments to the Author

Reviewer #1: The paper presents a motivated and well-designed proxy method for reducing the number of human rankings necessary to evaluate unpleasantness and detection time in electric vehicle sounds. Being able to reduce the search space for human subjects permits a significant reduction in the number of human trials necessary for an optimal evaluation, saving both time and money in the development process. The study in fact concludes that not only does their technique lessen cost and time, "the proposed heuristics lead to an improvement of the solutions found during the optimization process." The study concludes that modeling human responses fares better than finding an acoustical measurement derived from the literature. The results of this simulated study indicate that "A compromise should thus be found between the resources allocated to building the model and the risk of reducing the design space to an inadequate subspace." Arriving at that compromise would seem in itself to be a challenge in any real-world scenario that involves more variables and more subtle evaluations. The paper would be improved by some discussion of what kinds of problems, and what sorts of design spaces, might be most amenable to such a treatment. There is much value in their approach under the right circumstances, and for that reason I find this paper to be eminently worthy of publication. The extensive presentation and discussion of the methodology involved should help readers determine whether their method maps readily onto a different problem space, but I think the authors might assist by providing some consideration of that question in their conclusions. Tables 3 and 4 have some odd formatting in the headings of column three.

Reviewer #2: Authors in this research work have investigated the potential of using surrogate modeling for design space reduction, in the framework of interactive optimization for sound design purposes. Even though the method proposed in this paper is studied in the framework of electric vehicle sound design, it is generic enough to be applied to problems in other fields, for example fashion or user interface design. From this reviewer’s point of view, the topic and content of this paper were found interesting. The promising results have been achieved and evaluated in a well-organized manuscript, also the theoretical validation was provided. Although this paper seems attractive for readers, authors are requested to address the following comments to improve its quality prior to final recommendation.

1) The title is very general, please add more details to this section.

2) Abstract is too long, please summarize it more. The first two paragraphs can be deleted. This part can be supported with some numerical achievements as well. Advantages of the proposed work can be highlighted in this part.

3) Introduction section can be improved by adding more discussions on communications. For example, the antenna systems are very important for communications which can be mentioned in the introduction section along with proper references. Below are helpful suggestions.

“Dual-Polarized Highly Folded Bowtie Antenna with Slotted Self-Grounded Structure for Sub-6 GHz 5G Applications”, IEEE Transactions on Antennas and Propagation, vol. 70, no. 4, pp. 3028-3033, April 2022.

“New CRLH-Based Planar Slotted Antennas with Helical Inductors for Wireless Communication Systems, RF-Circuits and Microwave Devices at UHF-SHF Bands", Wireless Personal Communications- Springer Journal, Wireless Personal Communications, February 2017, Volume 92, Issue 3, pp 1029–1038.

“Hexa-Band Planar Antenna with Asymmetric Fork-Shaped Radiators for Multiband and Broadband Communication Applications” IET Microwaves, Antennas & Propagation, Volume 10, Issue 5, 13 April 2016, p. 471 – 478.

“Compact Rectifier Circuit Design For Harvesting GSM/900 Ambient Energy", Electronics, 2020, 9, 1614.

"Modified U-Shaped Resonator as Decoupling Structure in MIMO Antenna", Electronics, Volume 9, Issue 8, 1321, 2020.

"Impedance Bandwidth Improvement of a Planar Antenna Based on Metamaterial-Inspired T-Matching Network," IEEE Access, vol. 9, pp. 67916-67927, 2021.

4) Figure presents the summary of the proposed design space reduction strategies and experimental comparison protocol, please support this table with more elaborations.

5) Interesting equations have been provided in section 2, please explain how authors have extracted them?

6) Section 2 can be supported with some figures or plots to better understand the equations.

7) How are the results presented in Fig.6 obtained? Please discuss.

8) Please support the conclusion with some numerical findings.

6. PLOS authors have the option to publish the peer review history of their article (what does this mean?). If published, this will include your full peer review and any attached files.

Reviewer #1: No

Reviewer #2: No

---

## [Author Response · Author response to Decision Letter 0]

6 Nov 2023

Dear editor and reviewers, you will find attached with the revised manuscript, a response to your reviews and comments.

Best regards,

Mathieu Lagrange

---

## [Decision Letter · Decision Letter 1]

11 Dec 2023

Acoustical and Behavioral Heuristics for Fast Interactive Sound Design

PONE-D-23-15140R1

Dear Dr. Lagrange,

We’re pleased to inform you that your manuscript has been judged scientifically suitable for publication and will be formally accepted for publication once it meets all outstanding technical requirements. The reviewers are satisfied with the quality of your revision and believe that the manuscript quality has improved substantially. Please, look into the typo pointed out by Reviewer 1 before submitting your final files.

Within one week, you’ll receive an e-mail detailing the additional minor required amendments. When these have been addressed, you’ll receive a formal acceptance letter and your manuscript will be scheduled for publication.

Kind regards,

Sonsoles López-Pernas

Academic Editor

PLOS ONE

Reviewers' comments:

Reviewer's Responses to Questions

**Comments to the Author**

1. If the authors have adequately addressed your comments raised in a previous round of review and you feel that this manuscript is now acceptable for publication, you may indicate that here to bypass the “Comments to the Author” section, enter your conflict of interest statement in the “Confidential to Editor” section, and submit your "Accept" recommendation.

Reviewer #1: All comments have been addressed

Reviewer #2: All comments have been addressed

2. Is the manuscript technically sound, and do the data support the conclusions?

Reviewer #1: Yes

Reviewer #2: Yes

3. Has the statistical analysis been performed appropriately and rigorously? 

Reviewer #1: Yes

Reviewer #2: Yes

4. Have the authors made all data underlying the findings in their manuscript fully available?

Reviewer #1: Yes

Reviewer #2: Yes

5. Is the manuscript presented in an intelligible fashion and written in standard English?

Reviewer #1: Yes

Reviewer #2: Yes

6. Review Comments to the Author

Reviewer #1: This paper makes a valuable contribution and has addressed the concerns raised in the initial round of reviews. There remains, at least as far as I can see through the edits, a serious typo in the abstract which now states "We find that reducing by a factor of up to 64 an original design space of 4096 possible settings with the proposed heuristics reduces the number of iterations of the

design process by up to 2 to reach the same performance." which should make a claim of reducing the iterations of the design process by a power of up to 2, rather than 2. or, as is simply said in the conclusion "by a factor of two."

Reviewer #2: Authors have successfully addressed the reviewer's concerns. So, looking at the quality of the revised manuscript which shows a significant improvement than its initial version, there are no more technical comments from this reviewer's point of view.

7. PLOS authors have the option to publish the peer review history of their article (what does this mean?). If published, this will include your full peer review and any attached files.

Reviewer #1: No

Reviewer #2: No

---

## [Editor Report · Acceptance letter]

20 Dec 2023

PONE-D-23-15140R1 

PLOS ONE

Dear Dr. Lagrange, 

I'm pleased to inform you that your manuscript has been deemed suitable for publication in PLOS ONE. Congratulations! Your manuscript is now being handed over to our production team.

Kind regards, 

on behalf of

Dr Sonsoles López-Pernas 

Academic Editor

PLOS ONE